# EGFR in Cancer: Signaling Mechanisms, Drugs, and Acquired Resistance

**DOI:** 10.3390/cancers13112748

**Published:** 2021-06-01

**Authors:** Mary Luz Uribe, Ilaria Marrocco, Yosef Yarden

**Affiliations:** Department of Biological Regulation, Weizmann Institute of Science, Rehovot 7610001, Israel; mary.uribe@weizmann.ac.il (M.L.U.); ilaria.marrocco@weizmann.ac.il (I.M.)

**Keywords:** anti-cancer drug, drug resistance, growth factor, signal transduction, signaling pathway, transcription network, tyrosine kinase

## Abstract

**Simple Summary:**

Growth factors are hormone-like molecules able to promote division and migration of normal cells, but cancer captured the underlying mechanisms to unleash tumor growth and metastasis. Here we review the epidermal growth factor (EGF), which controls epithelial cells, the precursors of all carcinomas, and the cognate cell surface receptor, called EGFR. In addition to over-production of EGF and its family members in tumors, EGFR is similarly over-produced, and mutant hyper-active forms of EGFR are uniquely found in some brain, lung, and other cancers. After describing the biochemical mechanisms underlying the cancer-promoting actions of EGFR, we review some of the latest research discoveries and list all anti-cancer drugs specifically designed to block the EGFR’s biochemical pathway. We conclude by explaining why some patients with lung or colorectal cancer do not respond to the anti-EGFR therapies and why still other patients, who initially respond, become tolerant to the drugs.

**Abstract:**

The epidermal growth factor receptor (EGFR) has served as the founding member of the large family of growth factor receptors harboring intrinsic tyrosine kinase function. High abundance of EGFR and large internal deletions are frequently observed in brain tumors, whereas point mutations and small insertions within the kinase domain are common in lung cancer. For these reasons EGFR and its preferred heterodimer partner, HER2/ERBB2, became popular targets of anti-cancer therapies. Nevertheless, EGFR research keeps revealing unexpected observations, which are reviewed herein. Once activated by a ligand, EGFR initiates a time-dependent series of molecular switches comprising downregulation of a large cohort of microRNAs, up-regulation of newly synthesized mRNAs, and covalent protein modifications, collectively controlling phenotype-determining genes. In addition to microRNAs, long non-coding RNAs and circular RNAs play critical roles in EGFR signaling. Along with driver mutations, EGFR drives metastasis in many ways. Paracrine loops comprising tumor and stromal cells enable EGFR to fuel invasion across tissue barriers, survival of clusters of circulating tumor cells, as well as colonization of distant organs. We conclude by listing all clinically approved anti-cancer drugs targeting either EGFR or HER2. Because emergence of drug resistance is nearly inevitable, we discuss the major evasion mechanisms.

## 1. Introduction

Although numerous growth factors have been isolated and we are still debating the full spectrum of their physiological and pathological functions, there has been no question about the origin of growth factor research. Early studies performed in the 1930s in St. Louis University by Viktor Hamburger, one of the fathers of developmental neuroscience, assumed that the growth of nerve cells depended on an inductive agent emanating from their destination. To test this hypothesis, Hamburger removed developing limb buds from chick embryos and tested whether nerve cells near the spinal cord would still grow toward the limb. Because fewer embryonic nerve cells were observed when the limb buds had been removed, he concluded that nerve cells failed to grow because the limb was not producing an organizing factor on which they depended. Rita Levi-Montalcini, who repeated the experiments in Turin (Italy), and later received the Nobel Prize for the discovery of the Nerve Growth Factor (NGF), made similar observations. However, she reached a different conclusion. Studying the nerve cells at more frequent intervals, she noted that these cells did proliferate initially, but then died because the limb bud was not producing the putative growth factors needed for their survival. NGF was later isolated by Stanley Cohen, a biochemist who joined Levi-Montalcini and later shared the Nobel Prize with her, not before identifying both the second growth factor, the epidermal growth factor (EGF) [1] and the respective cell surface receptor (EGFR) [2]. Michael Waterfield and colleagues made the first link to cancer. They obtained partial amino acid sequences of human EGFR and discovered that this receptor very closely matches a part of the v-erb-B transforming protein of the avian erythroblastosis virus (AEV) [3]. Although subsequent studies found that EGFR and some ligand growth factors were overexpressed in human tumors and the *EGFR* gene is amplified or rearranged in brain tumors, frequent oncogenic point mutants of EGFR have been first identified in 2004, in human non-small cell lung cancer (NSCLC) from patients who were sensitive to EGFR-specific tyrosine kinase inhibitors (TKIs; e.g., gefitinib and erlotinib) [4,5,6].

This review will focus on EGFR (also called ERBB1 and HER1) and its seven growth factor ligands (EGF; transforming growth factor alpha, TGFa; heparin-binding EGF, HB-EGF; betacellulin; amphiregulin; epiregulin and epigen). In addition, we will refer to HER2/ERBB2, the closest family member, as well as to other EGFR family members. The reason for this is the ability of EGFR to recruit HER2, HER3 and HER4 in heterodimers, such that in some instances the functional unit is in fact an EGFR::HER2 or the other heterodimers. Notably, the EGFR family of receptor tyrosine kinases (RTKs) includes a catalytically defective member, HER3/ERBB3. In addition, unlike EGFR and both HER3 and HER4, which bind neuregulins (NRGs), HER2 binds with no known growth factor. This receptor acts as an amplifier of growth factor signals, in the context of a layered signaling network [7]. Each ERBB/HER receptor comprises a large extracellular domain (>500 amino acids), a single transmembrane domain, and a large cytoplasmic region harboring the catalytic tyrosine kinase activity. The reader is referred to similar reviews covering additional aspects of EGFR and the three other type I RTKs [8,9].

## 2. Physiological and Mutational Activation of EGFR

### 2.1. Ligand-Induced Receptor Activation

Under resting conditions, EGFR predominantly exists in an auto-inhibited monomeric form, but ligand binding confers a conformation poised to form either homodimers or heterodimers containing HER2 or other receptors. Activation of the receptor depends on the formation of an asymmetric dimer of kinase domains, in which one kinase domain allosterically activates the other [10]. In the asymmetric dimer, the kinase domain’s C-lobe of the “activator kinase” interacts with the N-lobe of the “receiver kinase”, which catalytically stimulates the latter. Thereafter, the receiver kinase trans-phosphorylates specific tyrosine residues of the activator kinase (Figure 1). The newly phosphorylated tyrosines serve as attachment sites for various adaptors (e.g., GRB2), cytoplasmic enzymes (e.g., PLC-gamma) or specific factors involved in transcription regulation (e.g., STAT3). Together, these signaling effectors and adaptor proteins link activated receptors directly or indirectly to canonical intracellular pathways, as well as to the endocytic machinery, which desensitizes active receptors. Importantly, several tyrosine docking sites of EGFR can bind more than one adaptor or effector. For example, phosphorylated tyrosine residue 1068 recruits the adaptor protein GRB2, which can initiate positive signaling pathways like the SOS-RAS route, or instigate negatively acting pathways, such as recruiting CBL, an E3 ubiquitin ligase that tags activated EGFRs with mono- or di-ubiquitins, to instigate EGFR sorting for degradation in lysosomes [11]. In addition, receptor connectivity is achieved by phosphorylation of cytoplasmic EGFR residues by intracellular kinases, such as SRC, which phosphorylates tyrosine 845, a residue that serves as an extra docking site. Altogether, this configuration permits the ligand-activated EGFR to simultaneously stimulate multiple intracellular signaling routes and gain robustness.

### 2.2. Mutation-Induced Receptor Activation

Oncogenic mutant forms of EGFR mimic the ligand-activated wild type form. Nevertheless, although the mutated EGFRs of tumors are enzymatically active and transforming, their tyrosine phosphorylation status is significantly lower compared to ligand-activated wild type receptors [12]. Apparently, this relatively low but persistent activity generates intracellular signals that differ from the canonical biochemical machinery. In addition, EGFR mutants frequently evade negative regulation (desensitization), such as receptor endocytosis and degradation [13], thus allowing the mutated receptors to act “under the radar” of receptor attenuating mechanisms.

#### 2.2.1. Extracellular Domain Mutations and Deletions

Glioblastoma multiforme (GBM) is the most common and deadliest primary brain tumor. Amplification of *EGFR* is a most prevalent aberration in GBM [14], as well as in carcinomas of the breast [15], colorectum [16], and lung [17]. In addition, an abundant EGFR deletion mutant has been identified (Figure 2; EGFRvIII). Likewise, several relatively rare point mutations in the EGFR’s extracellular and intracellular domains have been documented. EGFRvIII (also called EGFR∆III) represents the most frequent genetic aberration in brain tumors. Importantly, this and other EGFR mutations occur on background of *EGFR* amplification. EGFRvIII lacks large parts of the extracellular domain (ECD), and hence cannot bind with ligands, but it is constitutively active [18,19,20]. Although most abundant in glioblastomas [21,22,23], the EGFRvIII mutation also occurs in epithelial cancers, including breast, prostate, ovarian, and NSCLC. This mutant form of EGFR has been shown to be transforming both in vitro and in animal models. Furthermore, cells ectopically expressing EGFRvIII displayed reduced adhesion due to decreased focal adhesion size and number, as well as displayed enhanced migration [24]. EGFRs truncated at residue 958 and receptors lacking residues 521–603 are also common in GBM [25].

#### 2.2.2. Mutations within the Intracellular and Kinase Domains of EGFR

Lung cancer is the second most common malignancy and the leading cause of oncology-related deaths. A fraction of NSCLC patients presents activating mutations in the EGFR gene [4,5,6,26,27]. In similarity to brain tumors, lung tumors displaying EGFR mutations frequently associate with *EGFR* gene amplification [28]. In addition, the prevalence of the mutations varies in different human populations: approximately 10–12% of Caucasian patients with lung cancer versus up to 40% among East Asians. In fact, the discovery of these mutations accompanied efforts to understand why Asians, more than Caucasians, were highly sensitive to EGFR tyrosine kinase inhibitors (TKIs). As expected, in vitro assays and animal model studies have shown that mutant EGFRs are oncogenic [29,30]. The most common mutations are exon 19 deletions and an exon 21 point mutation, L858R [31] (Figure 2). In addition, relatively rare aberrations, such as point mutations, indels, and duplications occur within *EGFR*’s exons 18–25 [32]. Of the rare mutations, EGFR exon 20 insertions of 1–7 amino acids are the most common EGFR mutations in NSCLC, with frequencies ranging between 4–10% of all reported mutations. Erlotinib and gefitinib, the first-generation EGFR-specific TKIs, achieved clear superiority, in comparison to chemotherapy, in terms of patients’ progression free survival (PFS) [33,34,35,36]. However, despite initial good responses to the first-generation TKIs, patients inevitably become resistant within 10–14 months. As we will discuss below in more details, the most common mechanism of resistance entails secondary EGFR mutations (Table 1), especially emergence of the T790M mutation [37]. On-target mutations, along with bypass routes (e.g., *MET* and *HER2/ERBB2* amplification) bestow resistance to the 2nd generation TKIs [38,39]. Resistance to the third generation inhibitors involves additional mechanisms, including yet other on-target mutations [40], especially C797S [41,42]. The origin of recurring mutagenesis is attributable to the strong apoptosis-inducing effects of TKIs, which triggers down-regulation of mismatch repair and homologous recombination DNA-repair genes, along with concomitant up-regulation of error-prone DNA polymerases [43]. Additional mutant forms of EGFR, such as an EGFR fused with SEPT-14 (EGFR-SEPT14) [44,45], as well as EGFRs with a kinase domain duplication (EGFR-KDD) [46], have been identified in GBM, gastrointestinal and other tumors.

## 3. The Major Signaling Pathways Downstream of EGFR and Regulation of Transcription

One major pathway downstream of EGFR and other RTKs is the mitogen-activated protein kinase (MAPK) pathway [85]. Two adaptor proteins, GRB2 and SHC, link EGFRs to the ERK-MAPK pathway [86]. Both engage SOS, which stimulates RAS, and this results in activation of the RAF kinase (note that for simplicity RAF is not indicated in Figure 1). RAF proteins comprise the uppermost layer of a cascade of three kinases, which also includes MEK and the terminal MAPK, ERK [87]. In a similar manner, EGFR family members recruit a class I phosphatidylinositol 3-kinase (PI3K) [88]. Note, however, that EGFR harbors no direct PI3K docking site. PI3K phosphorylates phosphatidylinositol 4,5-bisphosphate (PIP_2_) to generate PI(3,4,5)P_3_ (PIP_3_). PIP_3_ recruits the AKT kinase to the plasma membrane. When bound to the inner leaflet of the plasma membrane, AKT undergoes activation by both PDK1 and mTORC2 (mTOR complex 2) [89]. AKT can inhibit apoptosis by means of phosphorylating BAD and FOXO family transcription factors. AKT can also activate mTOR by means of phosphorylating TSC2. Phosphorylation of TSC2 inhibits its GAP activity towards the GTPase RHEB. Active, GTP-bound RHEB proteins serve as activators of mTORC1, which controls both translation of mRNAs to proteins and the biosynthesis of cholesterol, which supplies lipids and proteins to growth factor stimulated cells [90,91].

The expression of approximately 1000 genes is affected following EGF stimulation of epithelial cells [92]. One major output of activated signaling pathways is phosphorylation or other changes in the transcription factors controlling this major response to growth factors. For example, the ELK-1 transcription factor is directly phosphorylated by ERK [93]. However, another transcription factor, CREB, is phosphorylated by a downstream kinase, p90RSK [94]. The first genes display increased expression approximately 10–20 min after challenging cells with a growth factor [95,96]. Transcripts corresponding to the most early genes, denoted immediate early genes (IEGs), rise rapidly after rising and then they fall. Importantly, the coordinated rise in IEG’s mRNA is preceded by rapid turnover of a class of 24 microRNAs, denoted immediately downregulated microRNAs (ID-miRs; see Figure 3) [97]. Transcription factors encoded by immediate early genes, such as FOS and EGR1, are included in the group of targets of ID-miR, raising the possibility that the observed high expression of ID-miRs in growth-arrested cells is aimed at silencing untimely IEG expression, thereby veto cell cycle entry. Unlike ID-miRs, a set of 22 microRNAs were induced by EGF prior to the initiation of mammary cell migration [98]. One member of this immediately upregulated microRNAs (IU-miRs) is miR-15b, which targets the 3’ untranslated region of MTSS1 (metastasis suppressor protein 1). In line with the tumor-suppressive actions of MTSS1, we found an inverse correlation between high expression of miRNA-15b and low expression of MTSS1 in tissues from breast cancer patients with the aggressive basal subtype, whereas low abundance of MTSS1 correlated with poor patient prognosis. In summary, coding and non-coding RNAs induced by EGF form a dense web of physical and functional interactions, which are strictly time-dependent.

Another set of genes, the delayed early genes, or DEGs [95], are activated approximately 2 h after the stimulus. Importantly, several genes within the DEG cluster act as suppressors of the IEGs. Congruently, several DEGs are consistently downregulated in a wide range of tumor types. A third set of genes begins to change approximately 2.5 h after stimulation with EGF. However, these genes display no further changes in expression and reach a steady state abundance between 4 and 8 h post-stimulation [99]. This delayed wave persists and determines long-term phenotype acquisition. For example, one of the newly synthesized molecules is Navigator-3 (NAV3), which regulates cell migration by decorating freshly polymerized microtubules and enhancing directional persistence of cell migration [100]. Analyses of breast and lung cancer patients associated low NAV3 with shorter survival, suggesting that by regulating microtubule dynamics and biasing directionally persistent rather than random migration, NAV3 inhibits locomotion of initiated cells. In a recent study, we identified multiple transcripts that are downregulated following prolonged treatment of mammary cells with EGF. Commonly, the delayed downregulated genes (DDGs) are lowly expressed in mammary tumors, and higher expression predicts better prognosis. One example is TSHZ2, a transcription factor downregulated upon EGFR stimulation. According to our yet unpublished results, TSHZ2 inhibits tumor growth and metastasis by nucleating a multi-protein complex regulating cytokinesis. Interestingly, existing methods for detecting expression dynamics often fail when the expression dynamics show a large heterogeneity. An alternative method, called TTCA (transcript time course analysis) has been designed for the analysis of perturbation responses [101]. This method captures both transient dynamics and slow expression changes, and its performance was tested on microarray data from EGF-stimulated cells. This method, when applied to a lung cancer model, uncovered EGF-induced elevation of interleukin 24, which might shift tumor cells from a migratory to a mitotic phenotype. In summary, time-dependent waves of EGFR-regulated mRNAs and non-coding RNAs (ncRNAs) establish molecular switches permitting control of the final steps of the cell division cycle, as well as the directionality of cell migration.

## 4. Crosstalk between EGFR and Other Signaling Pathways

In addition to direct mutational and non-mutational activation of EGFR, crosstalk pathways linking active EGFRs and other routes involved in cancer progression have been the subject of numerous studies. An example of this relates to the epigenetic EMT (epithelial-mesenchymal transition) program [102]. Although EMT has been broadly studied in the context of embryonic development, its involvement in cancer progression, metastasis, and EGFR signaling has gained burgeoning relevance in the last few years. Analysis of ovarian adenocarcinoma cells, which underwent EMT in response to treatment with EGF, unveiled regulation of both metabolism and cell cycle [103]. Interestingly, EMT did not affect cell proliferation rates, but led to cell cycle arrest regulated through increased levels of p21Waf1/Cip1, independently of p53. Another example relates to the crosstalk between EGFR and the Hippo/YAP pathway. This pathway is the main route regulating tissue enlargement and organ size, and it is activated in response to cell cycle arrest and downstream signaling to RAS [104]. Essentially, the Hippo pathway controls a kinase cascade that inactivates Yes-associated protein (YAP), which plays central roles in the progression of cervical cancer. According to a 2015 study, TGFa and amphiregulin, via EGFR, inhibit the Hippo signaling pathway and activate YAP, to induce cervical cancer cell proliferation and migration [105]. Furthermore, the human papilloma virus (HPV) E6 protein, a major etiological molecule of cervical cancer, maintains high YAP protein levels by preventing proteasome-dependent YAP degradation. Hence, the authors proposed that combined targeting of the Hippo and the EGFR pathways might constitute a novel therapeutic strategy for treatment of cervical cancer. Another pathway activated by EGFR stimulation is the NF-κB signaling cascade. Both EGFR- and NF-κB-dependent pathways establish positive loops that increase their oncogenic potential (reviewed in [106]). A 2016 study showed that MALT1 (mucosa-associated lymphoid tissue 1) is involved in EGFR-induced NF-κB activation in lung cancer cells, and that MALT1 inhibition impaired EGFR-dependent NF-κB activation [107]. Finally, it has been demonstrated that the long noncoding RNA (lncRNA) called *NEAT1* was regulated by EGFR signaling, and this was mediated by NF-κB [108].

Another crosstalk route enables synergistic interactions between EGFR and the hedgehog (HH) pathway [109]. Hedgehog ligands bind to the receptor, Patched (PTCH), which is localized in a nonmotile structure called ‘primary cilium’ [110]. Unliganded PTCH inhibits pathway activation by preventing entry of Smoothened (SMO) into the cilium, which results in the formation of repressor forms of the GLI transcription factor GLI3. Binding of HH to PTCH abolishes its repressive function on SMO. This allows SMO to enter the cilium and promote the formation of GLI activator transcription factor forms (GLI-A). In the absence of EGFR signals, GLI-A translocates to the nucleus and activates HH/GLI target genes. Simultaneous stimulation of HH/GLI and EGFR results in the synergistic activation of a group of genes, including SOX2, SOX9, JUN, CXCR4, and FGF19, thereby promoting malignant transformation in mouse models of HH/GLI driven basal cell carcinoma [111]. In conclusion, recent studies are resolving a dense network of cross-pathway interactions that likely coordinate EGFR signaling with activation of parallel pathways, thereby achieve robustness and coherence.

## 5. Roles for Non-Coding RNAs in EGFR Signaling

Epigenetic deregulation of gene expression is involved in the initiation and progression of multiple cancers. Herein we focus on the interesting interface connecting EGFR signaling and ncRNAs, such as the group of long ncRNAs (lncRNAs) and circular RNAs (circRNAs). Importantly, the surprising discovery that up to 90% of the human genome is subjected to pervasive transcription, although only less than 2% of the total genome encodes protein-coding genes, has placed ncRNAs in the limelight of the signal transduction and many other fields. As a result, we now understand that analyses of the approximately 7000 small RNAs, around 16,000 lncRNAs, and a slightly smaller number of pseudogenes [112] significantly changes our view of EGFR signaling.

For example, it has been reported that the levels of NEAT1, a lncRNA, were regulated by EGFR pathway activity, and this was critical for glioma cell growth and invasion [108]. Through binding to EZH2 and controlling the trimethylation of H3K27 in specific promoters, the EGFR/NEAT1/EZH2 axis contributes to glial cell tumorigenesis. Another lncRNA controlled by EGF stimuli is *LIMT* [113], long noncoding RNA inhibiting metastasis. Interestingly, LIMT downregulation required an active ERK pathway, and low expression of LIMT correlated with poor prognosis of patients with breast cancer. Furthermore, it was shown that knockdown of LIMT enhanced metastasis in an animal model of mammary gland tumors. Importantly, EGFR suppresses the expression of LIMT by enhancing histone deacetylation at the respective promoter. Such downregulation allows mammary cells to cross the extracellular matrix in vitro and enhance tumor metastasis in vivo. However, another lncRNA, TINCR, functions as a competing endogenous RNA (ceRNA) in human breast cancer, by means of upregulating EGFR expression and suppressing miR-503-5p expression [114]. In addition, TINCR enhances JAK2-STAT3 signaling downstream to EGFR, and STAT3 reciprocally increased the transcriptional expression of TINCR. These observations established a new STAT3-TINCR-EGFR feedback loop that might serve as a target for anti-cancer drugs.

In similarity to lncRNAs, microRNAs-mediated regulation has been shown to be involved in a wide range of biological processes, such as cell-cycle control, apoptosis, and several developmental and physiological mechanisms. In the context of EGFR signaling, it has been shown that the EGFR-to-MYC axis utilizes epigenetic mechanisms to silence microRNAs in gliomas [115]. The relevant microRNAs, miR-524-3p and miR-524-5p, were suppressed in glioblastoma and the suppression was associated with EGFR overexpression and the EGFRvIII mutation. Reciprocally, these two miRNAs associated with long overall survival time of patients with glioma. Likewise, another signaling axis, EGFR/miR-338-3p/EYA2, has been linked by a recent study to tumor growth and lung metastasis [116]. Accordingly, EGFR increases expression of the EYA2 oncoprotein by means of repressing tumor suppressor microRNA-338-3p (miR-338-3p) and activating the EYA2 oncoprotein. Remarkably, through the miR-338-3p/EYA2 pathway, EGFR increased breast cancer cell growth, EMT, migration, invasion, and metastasis in an allograft tumor mouse model [116].

Circular RNAs (circRNAs) are widespread circles of non-coding RNAs, which sequester microRNAs and RNA binding proteins, and, in some cases, contain short open reading frames [117]. However, although stimulation of epithelial cells with EGF leads to dynamic changes in the abundance of coding and non-coding RNA molecules, circRNAs display no similarly dynamic alterations [118]. In addition, these molecules are almost ubiquitously co-expressed with the corresponding linear transcripts, and due to their circular configuration they are, in general, long-lived [118]. Interestingly, it has recently been reported that a secretory E-cadherin protein variant (C-E-Cad), encoded by a circular E-cadherin RNA (circ-E-Cad), can directly activate EGFR [119]. Unexpectedly, C-E-Cad directly binds with the EGFR’s extracellular domain, thereby maintains glioma stem cell tumorigenicity. Presumably, due to its relatively high stability, circ-E-Cad undergoes multiple rounds of translation. In summary, recent reports are revealing intricate relations between EGFR signaling and several types of ncRNAs, both short (microRNAs) and long RNA molecules, such as lncRNAs and circRNAs. Future studies will weave together ncRNAs, mRNAs and proteins, thus reveal temporal and spatial interactions.

## 6. Proteomic and Epigenetic Analyses of EGF-Induced Molecular Switches

Along with transcriptional methods, proteome-based technologies uncover EGF-induced molecular switches. For example, combining SILAC (stable isotope labeling by amino acids in tissue culture), quantitative mass spectrometry and SUMO-IP (SUMOylated proteins immunoprecipitation), permitted the discovery of a switch instigated by small ubiquitin-related modifier (SUMO). The investigators identified the endogenous SUMOylated proteins of HeLa cells after EGF stimulation [120]. This uncovered a group of transcriptional coregulators, including IRF2BP1, IRF2BP2, and IRF2BPL, which are involved in the EGFR signaling pathway. Specifically, transient deSUMOylation of IRF2BP proteins was found to be essential for expression of the dual specificity phosphatase 1 (DUSP1) and the transcription factor ATF3. In a similar manner, phosphoproteomic analysis identified ladinin-1 (LAD1) as a phosphorylation-regulated mediator of the EGF-to-ERK pathway [121]. In response to EGF, LAD1 is transcriptionally induced, phosphorylated, and partly colocalizes with actin stress fibers, along with the actin cross-linking proteins called filamins. In line with enhancement of mammary cell growth and motility, LAD1 abundance predicts poorer patient prognosis and the corresponding transcript is highly expressed in aggressive subtypes of breast cancer.

RNA-sequence and chromatin immunoprecipitation (ChIP) sequencing for H3K18ac and H3K27ac (histone H3 lysine K18 and K27 acetylation) were used to find over 4000 modulated transcripts, including IEGs, DEGs and late-induced genes, after exposing HER2-overexpressing cells to EGF [122]. The data demonstrated that EGFR/HER2 signaling regulates the epigenome by means of time-dependent oscillation of global H3K18ac and H3K27ac following EGF treatment. These studies identified members of the S100 calcium binding protein as direct targets of EGFR signaling, since H3K18ac, H3K27ac, and RNA polymerase II (RNAPII) increase near the transcription start sites of these genes. A similar study examined the crosstalk between EGF and steroid hormones, which recurs in embryogenesis and is co-opted in cancer. When examined in mammary cells, the crosstalk demonstrated involvement of p53 and NF-κB, along with regulated pausing and traveling of RNAPII at the promoters and the bodies of EGF-inducible genes [123]. Essentially, the steroid hormone (glucocorticoid) inhibits positive feedback loops activated by EGF and stimulates the reciprocal inhibitory loops. In conclusion, relatively open (unmethylated) genomic regions controlling expression of feedback regulatory modules differentially recruit RNA polymerase II and acetylases/deacetylases to permit EGF-induced gene expression and enable the crosstalk between steroid hormones and growth factors.

## 7. Roles for EGFR in Metastasis

Although EGFR and its many ligands and co-receptors have been shown to be involved in multiple aspects of cancer progression [124], such as sustaining cell growth, enhancing resistance to cell death and reprogramming of metabolic networks, their involvement in metastasis has attracted notable attention in the last few years. Indeed, understanding the roles played by the many molecular players that fuel the metastasis cascade is vital for optimizing treatments of patients with advanced malignancies. This is due to the statistics showing that the overwhelming majority of cancer-associated deaths (>90%) are caused by metastatic disease, rather than the respective primary tumors [125,126]. Metastasis is initiated by disseminating tumor-derived cells, and it progresses through subsequent intravasation into blood vessels, culminating in colonization of distant organs. RTKs play critical roles in all steps necessary for the establishment of micro-metastases and secondary tumors. Studies performed in the last two decades have highlighted the regulation of metastasis by epigenetic and other reversible processes, such as EMT, chemotaxis and a plethora of reciprocal (paracrine) cell-to-cell interactions regulated by soluble factors [127]. In addition to initiating EMT, EGFR dynamically controls several actin-filled protrusions, which facilitate migration and invasion of cancer cells. They include lamellipodia and filopodia at the leading edge, and invadopodia facilitating invasion through the epithelium and basal membrane [128]. Actin filaments, along with RHO family GTPases [129,130,131], control not only cellular protrusions but also the switch from random to persistent migration [132].

Specifically, the RHO family GTPase called RAC promotes formation of peripheral lamellae, which mediate random migration [133], as well as contributes to persistent migration [131]. EGFR directly activates PLC-g1 by means of trans-phosphorylation. On the one hand, PLC-g1 generates gradients of diacylglycerol (DAG), which locally activate protein kinase C alpha, to guide chemotaxis [134]. On the other hand, PLC-g1 acts as a RAC1 guanine nucleotide exchange factor that boosts EGF-induced cell migration [135]. Additional guanine nucleotide exchange factors (GEFs) mediate the effects of EGFR and RAC1 on cancer cell migration. For example, the RAC-specific GEF called P-REX1 mediates actin cytoskeleton rearrangements and cell motility downstream to both EGFR and the G-protein coupled receptor CXCR4 [136]. Consistent with these observations, P-REX1 is highly expressed in estrogen receptor positive subtypes of breast cancer, and its abundance correlates with tumor spread in patients.

Invasion across tissue barriers requires cell softening. Unexpectedly, this step is preceded by transient accumulation of actin stress fibers and cell stiffening [137]. Invadopodia formation is regulated by EGF and other growth factors, as well as signals from the extracellular matrix [138,139]. The mechanism underlying invadopodia induction by EGF has recently been resolved [140]. EGF-activated cells are characterized by elevated levels of the phosphoinositol lipid PI(3,4,5)P_3_. Dephosphorylation of this lipid by synaptojanin 2 (SYNJ2) generates PI(3,4)P_2_, which recruits TKS5, an adaptor, to the plasma membrane, and nucleates invadopodia. Synaptojanin 2 is transcriptionally up-regulated on treatment of mammary cells with EGF. In addition, SYNJ2 is encoded at 6q25, a chromosomal locus amplified in aggressive forms of breast cancer. Normally, two tumor suppressor phosphatases, PTEN and INPP4B, deplete PI(3,4)P_2_ and balance the oncogenic alliance formed by PI3K and SYNJ2 [141]. In summary, by regulating inositol lipids and the actin cytoskeleton, growth factors can enhance invasiveness during both intravasation and extravasation. Alongside, EGFR enhances cancer cell invasiveness by harnessing the pro-invasive functions of TGFb [142]. Specifically, EGFR and ΔNp63, an isoform of the transcriptional regulator p63, potentiate TGFb induction of a subset of invasion-associated genes, along with transcriptional regulation of HBEGF, a heparin-binding EGFR ligand.

The intermediate state between primary and distant secondary tumors are circulating tumor cells (CTCs). According to a recent report, CTCs switch between proliferative and migratory states based on the dose of epigen, a low-affinity ligand of EGFR [143]. Both epigen and amphiregulin were highly induced upon tumor cell clustering, while depletion of epigen using RNA interference indicated that this ligand supports metastatic outgrowth from an internal microenvironment formed by CTC clusters. Evidence from prostate cancer similarly indicated that both EGFR and HER2 regulate metastasis to bone, the most prominent site of prostate cancer metastasis [144]. Specifically, EGFR may play a role in the survival of CTCs, whereas HER2 supports the growth of prostate cancer cells once they reached the metastatic sites. The final step of the metastatic cascade is the colonization of distant organs, a step dictated by reciprocal tumor-stromal interactions. Unlike prostate cancer, the peritoneum is the most common destination of ovarian cancer metastases. De novo expression of transforming growth factor-alpha (TGFa) is induced in omental stromal fibroblasts whereas tumor necrosis factor-alpha (TNFa) is expressed by ovarian cancer cells [145]. Apparently, TNFa induces TGFa transcription by stromal fibroblasts through nuclear factor-κB (NF-κB). In turn, fibroblast-made TGFa promotes peritoneal metastasis by means of activating EGFR signaling. Thus, a TNFa-TGFa-EGFR axis appears crucial for the peritoneal colonization of ovarian cancer cells. Presumably, additional paracrine loops between tumor cells and non-tumoral cells of the stroma propel cell migration/invasion and initiate the metastatic cascade. In line with this model, a previously reported paracrine loop involves EGF and colony-stimulating factor 1 (CSF-1; the macrophage growth factor). Accordingly, breast cancer cells expressing EGFR secrete CSF-1 and attract macrophages, which secrete EGF in the vicinity of cancer cells, thus permitting cancer cell migration [146].

## 8. Patient Resistance to Anti-Cancer Drugs Targeting EGFR

EGFR is one of the most successful pharmacological targets of anti-cancer drugs [147]. Both monoclonal antibodies (mAbs) and TKIs demonstrated efficacy and acceptable toxicity in large phase III clinical trials [8,34,35,36], hence were approved for treatment of lung, colorectal and head/neck cancer (see a list of anti-EGFR and anti-HER2 approved drugs in Table 1). However, despite major therapeutic advances, both primary and acquired resistance to these drugs occur and result in disease recurrence. Notably, while drug resistance arises from evolutionary pressures that select specific clones, resistance to TKIs often associates with appearance of new on-target mutations, but this mechanism rarely confers resistance to mAbs [38]. In addition, although the sequence of events preceding establishment of resistant clones is poorly understood, one commonality, which is shared by antibiotic-treated bacteria [148], entails an epigenetic transitory state, called drug tolerant persister (DTP) [149].

### 8.1. Resistance to EGFR-Specific TKIs

Both pre-existing and newly appearing on-target mutations drive the majority of resistance to first-generation EGFR TKIs [150]. The most common mechanism of resistance entails appearance of a secondary mutation (T790M) in the EGFR’s kinase domain [37]. This gatekeeper mutation causes drug resistance by increasing the affinity for ATP [151]. Other mechanisms of resistance to the first-generation TKIs include amplification of *MET* [47] or *HER2* [48], overexpression of *AXL* [49] or the hepatocyte growth factor (HGF) [50,51]. In addition, emergence of mutant forms of *RAS* [52] and *BRAF* [53], as well as phenotypic alterations [54], bestow resistance to the first-generation drugs. Second generation EGFR-specific TKIs were also developed. These inhibitors, called afatinib and dacomitinib, irreversibly bind with the ATP-binding clefts of the four EGFR/ERBB family members. Likewise, a third-generation TKI, osimertinib, which is able to irreversibly inhibit EGFR-T790M, has been developed [152]. Clinical trials comparing osimertinib to chemotherapy in T790M-positive NSCLC patients who failed a first line TKI, led to the approval, in 2015, of osimertinib as a second-line treatment [153]. Three years later, osimertinib was approved as a first-line treatment based on improved patient survival and reduced toxicity [154]. Recent data showed that the most common mechanisms of resistance to osimertinib in first-line settings are *MET* amplification, C797X mutations, which prevent covalent binding of the drug, amplification of wild-type *EGFR* or *HER2* and mutations in downstream signaling proteins [58]. Thus, despite the availability of several EGFR-specific TKIs, the long-term efficacy of these drugs is limited by multiple routes of acquired resistance.

### 8.2. Resistance to EGFR mAbs

The clinical approval of mAbs targeting EGFR for treatment of patients with metastatic colorectal cancer (mCRC) has represented a major step forward, primarily due to high efficacy in terms of progression-free survival and overall patient survival, along with improved quality of life [155,156]. However, the anti-EGFR mAbs were shown to be effective only in specific subsets of patients. Unlike resistance to TKIs, on-target alterations are rare and include an EGFR ectodomain mutation (S492R) [64]. Instead, several studies indicated that the major mechanisms underlying resistance to anti-EGFR antibodies converge into two pathways: RAS-RAF-ERK and PIK3K-AKT. Accordingly, the commonest mechanism of primary resistance of CRC to anti-EGFR antibodies involves genomic alterations affecting downstream effectors, such as *KRAS, NRAS,* and *PIK3CA* mutations. For instance, *KRAS* mutations in exon 2 (codons 12 and 13) were identified by several retrospective analyses as determinants of primary resistance to the antibodies [59,60]. Hence, patients with mutant forms of *KRAS* or *NRAS* are ineligible for treatment, since *RAS* mutations activate downstream pathways and establish a bypass survival route.

Unfortunately, even the responders eventually become resistant (secondary resistance) in less than 18 months [157]. In other words, among the patients with wild-type (WT) *KRAS* and *NRAS*, less than 35% respond to the combination of chemotherapy and anti-EGFR antibodies [158]. This is thought to be driven by mutational or epigenetic activation of pathways downstream of EGFR. Examples include *PTEN* deletions, *PIK3CA* mutations or *MET* activation. Along this line, two independent studies have shown that *HER2* amplification is a predictor of response to anti-EGFR antibodies [62,63]. Similarly, because EGFR is neutralized by anti-EGFR antibodies, the bypass route permitting resistance likely involves HER3 and its ligands, neuregulins. In analogy, several preclinical studies implicated the hepatocyte growth factor (HGF) and its receptor, MET, in resistance of CRC to anti-EGFR antibodies. Congruently, aberrant activation of MET may occur by means of either increased HGF expression or *MET* amplification. Interestingly, *MET* amplification was found in one of eight cetuximab-resistant specimens grown in mice (i.e., xenopatients) and bearing no mutations in *KRAS*, *BRAF*, *PIK3CA,* or *HER2* [61]. However, only extreme amplification of the *MET* locus has been associated with lack of response, which suggests that resistance is driven by a dosage effect. In conclusion, the molecular mechanisms that bestow primary (intrinsic) and secondary (acquired) resistance to EGFR inhibitors, including mAbs, are multiple and they share signaling attributes. These observations reinforce the roles played by tumor heterogeneity, pre-existing minor clones of cancer cells, and the adaptive mutability taking place while tumors are under treatment [43,159].

## 9. Conclusions

Because EGFR has been the pioneer member of the receptor tyrosine kinase family, and many lines of evidence linked this receptor, its growth factor ligands, and co-receptors to human malignancies, EGFR research has paved the way to several important concepts in cancer progression. Along with sustaining cell proliferation and conferring resistance to anti-cancer cytotoxic drugs, EGFR strongly promotes metastasis through a large collection of paracrine loops, each controlling a critical step of the metastasis cascade. In the last few years, we learned that diverse non-coding RNAs, including microRNAs, lncRNAs, and circular RNAs, play unexpectedly complex regulatory functions in EGFR signaling. Likewise, we have learned that the response to EGFR activation is surprisingly reproducible, highly ordered, and time-dependent: wave-like epigenetic events precede and license long-term commitments to specific cell fates. The newest results portray EGFR, along with its upstream and downstream interactors, as the axis of a layered network, which acts coherently with parallel networks and routes, such as NF-kB, hedgehog and the Hippo pathway. Congruent with the pivotal roles played by EGFR, the number of antibodies and kinase inhibitory drugs targeting this receptor and effectors is steadily increasing. Unfortunately, however, patients who initially respond to anti-EGFR drugs almost inevitably develop resistance. Resolving mechanisms of drug resistance translates to higher granularity maps of signaling pathways and it holds the promise of prolonging patient response.

## Figures and Tables

**Figure 1 cancers-13-02748-f001:**
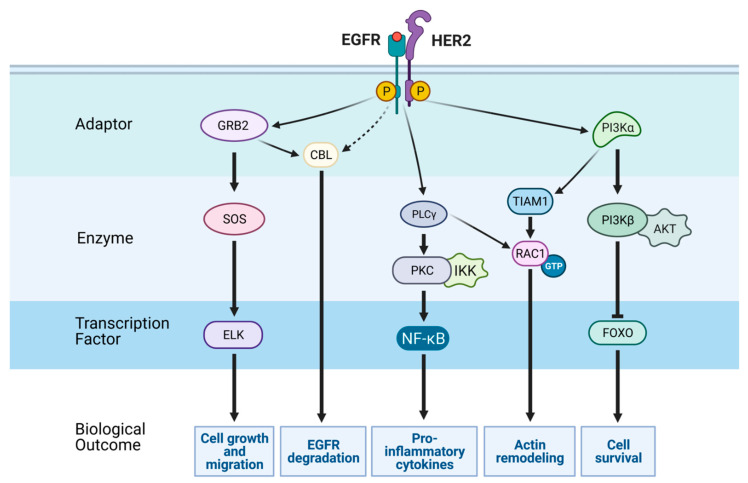
EGFR-mediated signaling pathways. Receptor tyrosine kinases (RTKs), such as EGFR, undergo rapid activation by means of growth factor binding, and subsequently they transmit signals through several pathways, to drive changes in cell phenotypes. Schematically shown are a heterodimer of EGFR and HER2, along with selected generic signaling routes comprising adaptors (green layer), enzymes (light blue layer), and transcription factors (blue layer). Note that CBL, an E3 ubiquitin ligase, is recruited directly or indirectly (via GRB2) to active EGFRs and sorts the latter to degradation in lysosomes. The active, GTP-bound form of RAC1 can be induced by several guanine nucleotide exchange factors (GEFs), including TIAM1 and phospholipase C-gamma, which acts as a GEF and a degrader of PI(4,5)P_2_. GRB2, growth factor receptor-bound protein 2; SOS, son of sevenless, CBL, Casitas B-lineage lymphoma; PLCg, phospholipase C gamma; PKC, protein kinase C; IKK, IκB kinase; NF-κB, nuclear factor κB; TIAM1, T-lymphoma invasion and metastasis-inducing protein 1; PI3K, phosphatidylinositol 3-kinase; AKT, AKR thymoma; FOXO, forkhead box O. This scheme was created using BioRender.com, accessed on 5 May 2021.

**Figure 2 cancers-13-02748-f002:**
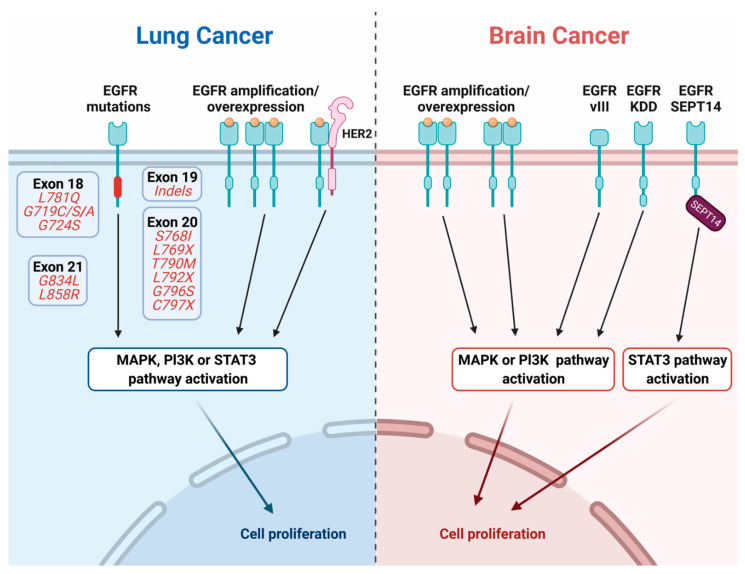
EGFR mutations and other main aberrations in lung and brain tumors. Point mutations and short deletions affecting the EGFR kinase domain are predominantly detected in NSCLC (left panel). The primary mutations are located in exon 19 or 21, while secondary mutations (post treatment) are located in exons 18, 20, and 21. In primary brain tumors, the most common mechanism of EGFR activation involves receptor overexpression and gene amplification. In addition, EGFR of brain tumors is affected by internal deletions that lead to the expression of truncation mutants. EGFRvIII is the major truncated form. It is associated with poor responses to conventional and EGFR-targeted therapies. Duplication of the kinase domain encoding sequence and a gene fusion are rare aberrations in brain tumors. This scheme was created using BioRender.com, accessed on 5 May 2021.

**Figure 3 cancers-13-02748-f003:**
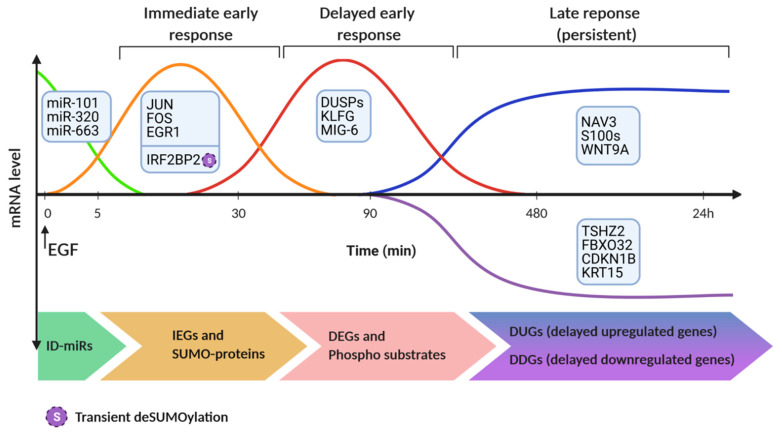
The timeline of transcription regulation and deSUMOylation following stimulation of EGFR. Immediately after the stimulus, a group of microRNAs (miRs), called ID-miRs, is downregulated. Next, the group of immediate-early genes (IEGs) undergoes up-regulation at the transcript level. Trailing the IEGs, the wave of delayed-early genes (DEGs), which encode cytokines, phosphatases and other inhibitors of growth factor signaling, undergoes up-regulation and switches-off the IEGs. Culminating the EGF-stimulated gene expression program, late-response genes display persistent up- (delayed upregulated genes, DUGs) or down-regulation (DDGs), and drive phenotypic changes. Note that posttranslational protein modifications, such as phosphorylation and deSUMOylation, occur in parallel to de-novo protein synthesis. This scheme was created using BioRender.com, accessed on 5 May 2021.

**Table 1 cancers-13-02748-t001:** Clinically approved kinase inhibitors and antibodies targeting EGFR and/or HER2/ERBB2.

Receptor	Class	Name of the Drug	Trade Name	Year of Approval	Company	Indication	Main Resistance Mechanisms
**EGFR**	TKI	Gefitinib	Iressa	2003	AstraZeneca	NSCLC	*Resistance to first and second generation EGFR-TKIs*: T790M mutation [37], amplification of MET [47] or HER2 [48], overexpression of AXL [49] or HGF [50,51], mutations in downstream molecules [52,53], phenotypic transformation [54]. *Resistance to 2nd-line osimertinib*: C797S mutation [42], MET or HER2 amplification [55], activation of downstream signaling pathways [56], phenotypic transformation [57]. *Resistance to 1st-line osimertinib:* MET amplification, C797X mutations, EGFR WT or HER2 amplification, mutations in downstream molecules [58].
		Erlotinib	Tarceva	2004	Genentech/OSI	NSCLC/Pancreatic cancer	
		Afatinib ^#^	Gilotrif	2013	Boehringer Ingelheim	NSCLC	
		Dacomitinib ^#^	Vizimpro	2018	Pfizer Inc.	NSCLC	
		Osimertinib	Tagrisso	2015	AstraZeneca	NSCLC	
		Vandetanib *	Caprelsa	2011	Genzyme Corp	Medullary thyroid cancer	
	mAb	Cetuximab	Erbitux	2004	Imclone/Bristol-Meyers Squib	CCR, head and neck cancer	*Resistance to EGFR mAbs*: Genomic alterations in downstream molecules [59,60], activation of bypass receptors (e.g., MET, HER2) [61,62,63], mutations in the extracellular domain of EGFR (e.g., S492R) [64].
		Panitumumab	Vectibix	2006	Amgen	CCR	
		Necitumumab	Portrazza	2015	Eli Lilly Co	NSCLC	
**HER2**	TKI	Lapatinib	Tykerb	2007	GSK	Breast cancer	*Resistance to HER2 TKIs*: Inhibition of apoptotic pathways [65,66], activation of bypass receptors (i.e., HER3, MET, AXL, ER) [67,68,69,70], activation of downstream signaling pathways [71,72], T798I mutation [73,74].
		Neratinib ^#^	Nerlynx	2017	Puma Biotech	Breast cancer	
		Tucatinib	Tukysa	2020	Seagen	Breast cancer	
	mAb/ADC	Trastuzumab	Herceptin	1998	Genentech	Breast cancer, gastric or GEJ adenocarcinoma	*Resistance to HER2 mAbs*: Expression of a truncated form of HER2 (p95HER2) [75], decreased mAb binding to the receptor due to overexpression of Mucin 4 (epitope masking) [76], activation of downstream signaling pathways [77,78,79], activation of bypass receptors [80,81,82], impaired ADCC (FcγRIII 158 V/F or 158 F/F polymorphisms) [83], loss of HER2 [84].
		Pertuzumab	Perjeta	2012	Genentech	Breast cancer	
		Trastuzumab emtansine	Kadcyla	2013	Genentech	Breast cancer	
		Trastuzumab deruxtecan	Enhertu	2019	Daiichi Sankyo	Breast cancer, gastric or GEJ adenocarcinoma	
		Margetuximab	Margenza	2020	Macrogenics Inc	Breast cancer	

ADC, antibody drug conjugate; ADCC, antibody-dependent cellular cytotoxicity; CRC, colorectal cancer; GEJ, gastroesophageal junction; mAb, monoclonal antibody; NSCLC, non-small cell lung cancer; TKI, tyrosine kinase inhibitor. * It also inhibits VEGFR and RET. ^#^ Pan-HER inhibitor.

## Data Availability

No new data were created or analyzed in this study. Data sharing is not applicable to this article.

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
