# Peer review of "EGFR in Cancer: Signaling Mechanisms, Drugs, and Acquired Resistance"

_cancers, 2021, doi:10.3390/cancers13112748_

Round 1

Reviewer 1 Report

The submitted manuscript has reviewed the mechanisms of EGFR activation and its downstream signaling involving transcriptional factors, non-coding RNA, and proteomic and epigenetic molecules.

Overall, the manuscript is convincing and comprehensive. While some issues need clarifications, there is sufficient information in this manuscript to warrant its publication.

Comments:

Although it is a well-written review, the manuscript lacks a section on Conclusion.

The authors compared the EGFR activation and related signaling molecules between wild-type and mutated proteins in sections 2.1 and 2.2. They should clearly illustrate the differences in the EGFR activation signals in the two proteins especially at the adaptor or enzyme levels in Figure 1.

Reviewer 2 Report

In this review, the authors briefly introduced the discovery stories of EGFR and background information of EGFR family members. The authors detailed physiological and ectopic activation of EGFR, EGFR activated signaling pathways and its dynamicity, relationship between EGFR and other pathways, ncRNAs and its associated -omics changes, roles of EGFR in metastasis and drug resistance. Given the current clinical significance of EGFR inhibitors, this is a timely review with comprehensive elucidation. The following comments may strength the MS:

  1. Except for activation mutation, gene amplification of EGFR is observed in many cancers. The authors might as well include it in part 2.
  2. In part 2.2, the authors delineated activation mutation of EGFR in two separate parts based on mutation sites on EGFR – extracellular domain and kinase domain/cytoplasmic region. However, the EGFR-KDD and EGFR-SEPT14 in line 129 are the mutations at cytoplasmic region and therefore it might be inappropriate to include in part 2.2.1 with a header “Extracellular domain mutations and deletions”.
  3. In Figure 3, the authors presented dynamic response of microRNAs and mRNAs upon EGF stimulation. Accordingly, the authors detailed the process of downregulation of ID-miR, immediate early response and delayed early response in part 3. However, late response in figure 3 seems to be left unmentioned in part 3.
  4. In part 4, the authors listed a couple of pathways activated by EGFR. To fill in the “crosstalk” map, other pathways that are downregulated by EGFR signaling or important signals that activate/inhibit EGFR signaling should also be included in this part if possible.
  5. In part 7, the authors emphasized the role of EGFR in metastasis. EGFR promotes tumor malignancy in many other aspects as well, such as unlimited tumor growth, resistance to cell death, immune evasion, metabolic reprogramming, etc. It I suggested to consider collectively covering these points before highlighting its role in metastasis.
  6. In line 492-493, the authors mentioned that anti-EGFR mAbs were only effective in certain subsets of patients. The authors might further characterize these groups of patients before discussing the emerging resistant mechanisms.
